# BK channel inhibition by strong extracellular acidification

Yu Zhou*, Xiao-Ming Xia, Christopher J Lingle

Department of Anesthesiology, Washington University School of Medicine, St. Louis, United States

**Abstract** Mammalian BK-type voltage- and $Ca^{2+}$-dependent $K^+$ channels are found in a wide range of cells and intracellular organelles. Among different loci, the composition of the extracellular microenvironment, including pH, may differ substantially. For example, it has been reported that BK channels are expressed in lysosomes with their extracellular side facing the strongly acidified lysosomal lumen (pH ~4.5). Here we show that BK activation is strongly and reversibly inhibited by extracellular $H^+$, with its conductance-voltage relationship shifted by more than +100 mV at $pH_O$ 4. Our results reveal that this inhibition is mainly caused by $H^+$ inhibition of BK voltage-sensor (VSD) activation through three acidic residues on the extracellular side of BK VSD. Given that these key residues (D133, D147, D153) are highly conserved among members in the voltage-dependent cation channel superfamily, the mechanism underlying BK inhibition by extracellular acidification might also be applicable to other members in the family.

DOI: https://doi.org/10.7554/eLife.38060.001

## Introduction

A hallmark of BK-type $Ca^{2+}$ and voltage-activated $K^+$ channels is its dual regulation by two distinct physiological signals, $Ca^{2+}$ and membrane potential (*Latorre et al., 1982*; *DiChiara and Reinhart, 1995*; *Horrigan and Aldrich, 2002*). The key $Ca^{2+}$ and voltage-sensing functions of the BK channel are embedded within the pore-forming BK α subunit and arise from interactions among three linked modules: a central pore-gate domain (PGD) that forms a $K^+$-selective ion permeation pathway, a transmembrane voltage-sensor domain (VSD) similar to that of the voltage-gate $K^+$ (Kv) channels (*Ma et al., 2006*), and two regulator of conductance for $K^+$ (RCK) domains that form a cytosolic gating ring (CTD) to provide binding sites for intracellular ligands such as $Ca^{2+}$ (*Schreiber and Salkoff, 1997*; *Jiang et al., 2001*; *Shi et al., 2002*; *Xia et al., 2002*, *2004*). Yet, despite the common $Ca^{2+}$- and voltage-sensing functions of all BK channels, BK channels typically exhibit considerable diversity in their functional properties among different loci of expression, for example, in epithelial cells (*Bailey et al., 2006*; *Grimm et al., 2007*), smooth muscle (*Robitaille et al., 1993*; *Wellman and Nelson, 2003*; *Tanaka et al., 2004*), neurons (*Reinhart et al., 1991*; *Rosenblatt et al., 1997*; *Meredith et al., 2006*), potentially even in various organelles (*Singh et al., 2013*; *Li et al., 2014*; *Cao et al., 2015*; *Wang et al., 2017*), thereby allowing them to play physiological roles suited to their particular location of expression. This diversity in function occurs despite the fact that only a single gene (KCa1.1; Kcnma1) encodes the pore-forming subunit of all BK channels. As a consequence, a critical part of understanding potential physiological roles of BK channels has been in uncovering the various factors and mechanisms that define cell-specific functional properties.

What factors contribute to defining the physiological roles of BK channels? At least some diversity in BK channel properties can arise from the stable cell-specific molecular composition of the BK channel complex, including alternative splicing of pore-forming α subunits (*Tseng-Crank et al., 1994*; *Rosenblatt et al., 1997*) and co-assembly with various accessory subunits (*Wallner et al., 1996*; *Xia et al., 1999*; *Brenner et al., 2000*; *Xia et al., 2000*; *Brenner et al., 2005*; *Yan and*

*For correspondence:
zhouy@wustl.edu

Competing interests: The authors declare that no competing interests exist.

*Aldrich, 2010*). BK channels can also be regulated on a shorter time scale by post-translational modifications such as phosphorylation (*Tian et al., 2004, 2008*) or even more briefly by binding of soluble factors to the BK channel complex, such as $Ca^{2+}$ (*Magleby and Pallotta, 1983*; *Schreiber and Salkoff, 1997*; *Xia et al., 2002*), $Mg^{2+}$(*Shi and Cui, 2001*; *Zhang et al., 2001b*; *Shi et al., 2002*; *Horrigan and Ma, 2008*), heme (*Tang et al., 2003*; *Horrigan et al., 2005*), carbon monoxide (*Hou et al., 2008b*), or omega-3 fatty acid (*Hoshi et al., 2013b, 2013a*). In the latter cases, modulation of BK activity appears to involve interactions with the regulatory factors to binding sites on the intracellular side of the BK channel. In contrast, despite the fact that the extracellular side of BK channels may be exposed to a wide range of microenvironments, little is known about how and even whether regulation of BK channels by extracellular soluble factors may occur.

One common potential extracellular regulator might be protons ($H^+$). Many BK channels are expressed in loci where they may experience extremely low extracellular pH ($pH_O$ <5). For example, BK channels regulate potassium secretion in the distal nephron (*Bailey et al., 2006*; *Grimm et al., 2007*), while urinary pH ranges from 6.5 to 7.5 daily in healthy people and may vary from 4.5 to 8.0 in pathological conditions (*Wagner et al., 2004*). Recently, BK channels were proposed to regulate $Ca^{2+}$ release/refilling and membrane trafficking of lysosomes (*Cao et al., 2015*; *Wang et al., 2017*). In the case of lysosomes, the BK channels are oriented with their extracellular side exposed to a high concentration of $H^+$ in the lumen of the lysosome, the pH of which ranges from 4.2 to 4.7 (*Mindell, 2012*; *Xu and Ren, 2015*). Some previous work on Kv1 (*Claydon et al., 2000*; *Kwan et al., 2006*; *Claydon et al., 2007*, *Claydon et al., 2008*) and KCNQ channels (*Prole et al., 2003*) establishes precedence that some Kv channels may be sensitive to $pH_O$ in the pH range of 5–7, perhaps by effects on the pore region of these channels (*Zhang et al., 2003*; *Claydon et al., 2008*; *Wang et al., 2017*). However, the impact of $pH_O$ below pH five such as that which might be found in the lumen of a lysosome has not been systematically examined in Kv channels.

In this study, we address the questions of whether BK channels are sensitive to changes in $pH_O$ and what the underlying mechanism of $pH_O$ regulation might be. We report a strong and reversible inhibition of BK channel function by extracellular $H^+$ beginning around $pH_O$ 6. The primary effect of low $pH_O$ on BK channel function is to inhibit the activation of the BK VSD. In addition, a moderate change in the closed-open equilibrium of the BK permeation gate in favor of closed states also contributes to inhibition induced by extracellular acidification. The combination of these two effects positively shifts the BK conductance-voltage (G-V) relationship by more than 100 mV at $pH_O$ 4. We further show that three acidic residues on the extracellular side of the BK VSD are involved in such inhibition. As stated above, BK channels are widely expressed on plasma membranes of a diverse variety of cells, some of which may be exposed to unusual and varying extracellular milieu, for example, in the kidney (*Bailey et al., 2006*; *Grimm et al., 2007*) or intestinal crypts (*Sandle et al., 2007*; *Sørensen et al., 2011*). Furthermore, BK channels have been reported to be in the membranes of intracellular organelles (*Cao et al., 2015*; *Wang et al., 2017*), in which the normally extracellular side of the BK channel will face the organelle lumen, often a highly acidic compartment. Given such factors, the present results impact on the proper assessment of the potential physiological and pathological roles that BK channels may play in such loci. Furthermore, given that the three key residues identified in this study are highly conserved among members in the voltage-dependent cation channel superfamily, the mechanism underlying BK inhibition by strong extracellular acidification might also be applicable to Kv channels and other members in this family, some of which also perform their physiological function in extremely acidic environments (*Heitzmann and Warth, 2007*).

## Results

### BK channel activation is strongly inhibited at $pH_O$ lower than 5

Given the possibility that BK channels may be expressed in loci in which the normally extracellular side of BK channels is exposed to acidic conditions, here we have examined the impact of $pH_O$ on BK channel function.

*Figure 1A* shows macroscopic BK currents recorded from an outside-out patch perfusing in extracellular solutions at pH 7 (left), pH 4 (middle), and back to pH 7 (right). The amplitude of macroscopic BK current was reduced at $pH_O$ 4, with reduction became increasingly apparent as the voltage is made more negative, for example, comparing moderate depolarization at +60 mV

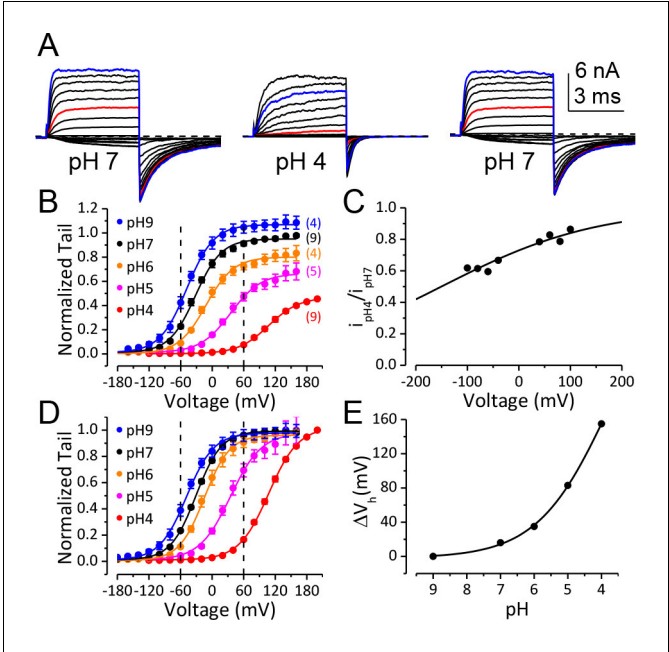

**Figure 1.** BK channel activation is strongly inhibited when $pH_O$ is lower than 5. (**A**) Macroscopic BK currents recorded from an outside-out patch perfused with extracellular solutions at pH 7 (left), pH 4 (middle), and back to pH 7. Currents were evoked by steps from −160 to +160 mV at $pH_O$ 7 and by steps from −160 mV to +200 mV at $pH_O$ 4 (20 mV increments). The pre- and post-test pulse voltages were −140 and −120 mV, respectively. 300 µM $Ca^{2+}$ was included in pipette (intracellular) solution. The red traces were evoked by +60 mV pulses to highlight the much larger fractional reduction in outward current at this voltage compared to that at +160 mV (blue traces). (**B**) G-V curves of BK channel generated from tail currents recorded at various $pH_O$s with values normalized to the maximum value at $pH_O$ 7. Physiologically relevant voltage range (±60 mV) is marked by two vertical dotted lines here and in panel D. Boltzmann fit results (solid lines) are $G_{max}$ = 1.06 ± 0.01, z = 0.97 ± 0.03 e, Vh = −48.8 ± 1.0 mV (pH 9), $G_{max}$ = 0.95 ± 0.01, z = 1.00 ± 0.04 e, Vh = −32.4 ± 1.2 mV (pH 7), $G_{max}$ = 0.80 ± 0.01, z = 0.97 ± 0.06 e, Vh = −11.7 ± 1.8 mV (pH 6), $G_{max}$ = 0.66 ± 0.01, z = 0.86 ± 0.05 e, Vh =+34.8 ± 2.3 mV (pH 5), $G_{max}$ = 0.46 ± 0.02, z = 0.90 ± 0.09 e, Vh =+107 ± 3.7 mV (pH 4). The number in parentheses is the number of experiments contributing to each G-V relationship. (**C**) Voltage-dependent change of BK single channel current induced by extracellular acidification. The filled dots are fractional BK single channel current amplitude determined over the voltage range of ±100 mV when $pH_O$ changed from 7 to 4. The Woodhull model (refer to Materials and methods) fit results (solid line) are: $k_{d0}$ = 0.27 ± 0.02 mM (pH 3.6) and $\delta$ = 0.17 e ± 0.02. The prediction of fractional BK single channel current amplitude by the Woodhull model is extended to ±200 mV. Note that the residual single channel current at −120 mV at $pH_O$ 4.0 is predicted to be 0.55 of that at $pH_O$ 7.0, which is comparable to the $G_{max}$ at $pH_O$ 4 in panel B. (**D**) BK G-V curves as those shown in panel B, but with the values at each $pH_O$ normalized to the maximum value within that $pH_O$. Boltzmann fit results (solid lines) are $G_{max}$ = 0.97 ± 0.01, z = 0.98 ± 0.04 e, $V_h$ = −48.3 ± 1.1 mV (pH 9), $G_{max}$ = 0.98 ± 0.01, z = 1.05 ± 0.04 e, $V_h$ = −30.5 ± 1.1 mV (pH 7), $G_{max}$ = 0.96 ± 0.01, z = 1.04 ± 0.05 e, $V_h$ = −14.5 ± 1.2 mV (pH 6), $G_{max}$ = 0.97 ± 0.01, z = 0.86 ± 0.04 e, $V_h$ =+35.0 ± 1.6 mV (pH 5), $G_{max}$ = 1.01 ± 0.02, z = 0.90 ± 0.04 e, $V_h$ =+107.6 ± 1.8 mV (pH 4). The $V_h$ and z values are virtually identical to those determined from the G-Vs in *Figure 1B*. (**E**) Dose-response curve of BK gating shift ($\Delta V_h$) induced by extracellular $H^+$. Hill equation ($\Delta V_h = \Delta V_{max}/(1+(IC_{50}/[H^+])^n)$) fit result (solid line) is: $IC_{50}$ = 0.17 ± 0.28 mM (pH 3.8), n = 0.41 ± 0.07.

DOI: https://doi.org/10.7554/eLife.38060.002

The following figure supplement is available for figure 1:

**Figure supplement 1.** BK channel single channel current at various voltages and $pH_O$s.

DOI: https://doi.org/10.7554/eLife.38060.003

(*Figure 1A*, red traces) to that at +160 mV (*Figure 1C*, blue trace), or comparing the reduction in inward tails at −120 mV (*Figure 1A and B*) to the reduction in outward current at +160 mV (*Figure 1C*, blue trace). A previous study showed that neutralization of a highly conserved acidic residue (D292N) in the BK channel pore region reduces single channel conductance and induces an outward rectification in resulting BK channels (*Haug et al., 2004b*). In a similar fashion, we observed

that BK single channel conductance was reduced by ~50% at −120 mV and ~20% at +60 mV when pH$_O$ was changed from 7 to 4 (*Figure 1C*, *Figure 1—figure supplement 1*). This change in single channel conductance can largely account for the reduction of maximum tail current at −120 mV (*Figure 1B*). However, it is also clear that the reduction of BK single channel conductance induced by extracellular acidification accounts for only a small portion of the changes in outward current amplitude over the physiologically relevant voltage range (e.g. the amplitude of macroscopic BK current at +60 mV was reduced by ~85% when pH$_O$ was changed from 7 to 4). At more positive potentials, the amplitude of BK current at pH$_O$ 4 was closer to that at pH$_O$ 7. However, the reduction in macroscopic current amplitude with pH$_O$ changed from 7 to 4 was greater than that in single channel conductance over all the voltages we examined. For example, the amplitude of current evoked by +160 mV pulse at pH$_O$ 4 was only 0.72 ± 0.02 (n = 9) of that at pH$_O$ 7, while the BK single channel conductance at pH$_O$ 4 was estimated to be 0.88 of that at pH$_O$ 7 at the same voltage (*Figure 1C*). Above results indicate that extracellular acidification strongly shifts the voltage-dependent gating of BK channel toward positive potential.

We therefore plotted the conductance-voltage (G-V) relationship for BK activation at a given pH$_O$ normalized to the maximum conductance at that pH$_O$ (*Figure 1D*). This revealed that the G-V relationship was shifted toward positive potentials with increases in extracellular [H$^+$]. The BK half-activation potential (V$_h$) was shifted by more than +60 mV when pH$_O$ was changed from 7 to 5 and was further shifted by another +70 mV at pH$_O$ 4. With a gating shift of this magnitude, the P$_O$ of BK did not reach maximum level even at +160 mV when pH$_O$ was changed from 7 to 4 (*Figure 1D*).

In addition to reduction in current amplitude, the kinetics of BK current was also significantly changed with elevation of extracellular [H$^+$], with activation being much slower while deactivation being much faster at pH$_O$ 4 than at pH$_O$ 7 (*Figure 1A*). These changes in channel kinetics are also consistent with the gating of BK channels shifted toward positive potentials by extracellular H$^+$. Importantly, the activation of BK channels remained low over the physiologically relevant voltage range of ±60 mV at pH$_O$ 4 even with saturating [Ca$^{2+}$]$_{in}$ (*Figure 1D*). Therefore, the primary contributor to reduction in BK activation at pH$_O$ 4.0 over these voltages is the shifted BK gating but not the reduction in single channel conductance. The gating shift induced by extracellular H$^+$ appeared to be independent of [Ca$^{2+}$]$_{in}$ as a comparable G-V shift was observed with 10 μM [Ca$^{2+}$]$_{in}$ (Figure 3C). It should be noted that even with a rather modest gating shift such as that induced by pH$_O$ of 6 (~20 mV), the resulting reduction in the BK channel current could still be sufficient to markedly alter the physiological function of BK channels.

The change of V$_h$ induced by extracellular acidification is plotted against pH$_O$ in *Figure 1E*. The approximate IC$_{50}$ of pH$_O$-dependent BK G-V shift is pH 3.8, suggesting that acidic residue(s) on the extracellular side of the BK channel may be involved in this inhibition.

## A change in the C-O equilibrium only accounts for a small portion of the gating shift induced by extracellular H$^+$

We next sought to address the question of the mechanistic basis for the rightward gating shift induced by extracellular protons. Extracellular H$^+$ might inhibit BK channel activation by several different mechanisms, including inhibition of either opening of the BK permeation gate or activation of the BK VSD. The modular design of BK channel protein and allosteric nature of BK channel gating make it possible to examine these possibilities separately. Previous studies suggest that extracellular acidification inhibits Kv1 channels by enhancing inactivation in the pore domain of these channels (*Claydon et al., 2007*; *Kwan et al., 2006*; *Trapani and Korn, 2003*; *Zhang et al., 2003*). Therefore, we started by examining the effects of extracellular acidification on the closed-open equilibrium in the BK pore region. To do this, we recorded BK single channel currents at negative potentials (−100, −120, −140 mV) where BK VSDs are largely in resting states even with elevated concentrations of [Ca$^{2+}$]$_{in}$ up to 100 μM (*Horrigan and Aldrich, 2002*; *Carrasquel-Ursulaez et al., 2015*). A change in limiting open probability times the number of BK channels (limiting nP$_O$) determined from such recordings reflects a change in the C-O equilibrium of the BK pore gate (*Horrigan et al., 1999b*). As shown in *Figure 2A and B*, the limiting nP$_O$ of BK channels was moderately reduced when pH$_O$ was changed from 7 to 4. The average limiting nP$_O$ of BK channels at pH$_O$ 4 was between 0.3 and 0.4 of that at pH$_O$seven among the voltages we tested, indicating that the C-O equilibrium constant L of BK channels at pH$_O$ 4 was about 1/3 of that at pH$_O$ 7. To evaluate the impact of such reduction of L on BK channel gating we simulated G-V relationships using the Horrigan-Aldrich (H-A)

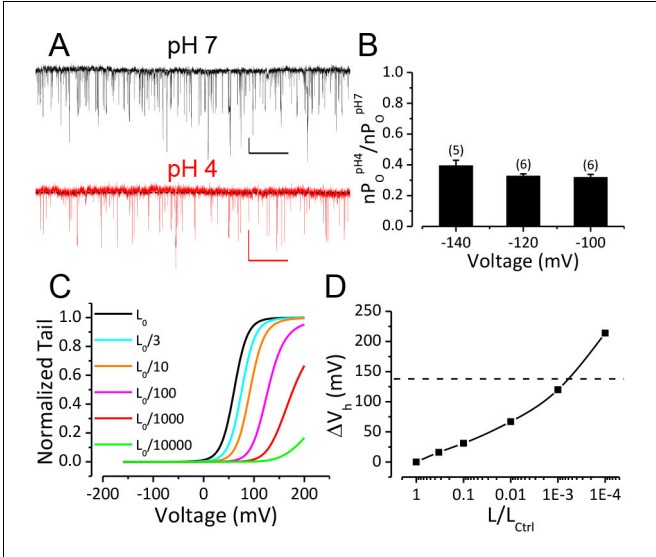

**Figure 2.** Reduction of BK C-O equilibrium constant $L$ only accounts for a small portion of observed gating shift by extracellular acidification. (**A**) BK single channel activity recorded from an outside-out patch held at −100 mV perfusing in solutions at pH 7 or pH 4. 10 μM $Ca^{2+}$ was included in pipette solution. Opening is downward. The $nP_O$ of the trace at pH 7 is 0.067. The $nP_O$ of the trace at pH 4 is 0.03. Scale bars: 10 pA vertical, 50 ms horizontal. (**B**) Fractional $nP_O$ of BK channels when $pH_O$ is changed from 7 to 4 at three negative holding potentials (−100, −120, −140 mV). The number of patches obtained at each potential is listed above each column. There is no significant difference among values determined at these three holding potentials ($p > 0.05$, one-way ANOVA). (**C**) G-V curves calculated using the H-A model with published parameters: $K_d = 11$ μM, C = 8, D = 24, E = 2.4, $L_0 = 10^{-6}$, $V_{hC} = 156$ mV, $z_j = 0.58$ e, $z_l = 0.3$ e (**Horrigan and Aldrich, 2002**) to highlight the potential impact of changes in $L_0$ (zero voltage C-O equilibrium constant). $L_0$ is scaled down from the published value of $10^{-6}$ (black line, $L_{0ctrl}$) to shift the G-V toward positive potentials. $V_h$ is shifted by +16 mV when $L_0$ is reduced to $0.33 \times 10^{-6}$ (cyan line, $L_{0ctrl}/3$). (**D**) Change of $V_h$ plotted against fold-reduction in $L_0$. The data points are connected by a cubic spline line with no physical meaning. Dotted line marks the change of $V_h$ observed in our experiment when $pH_O$ was changed from 7 to 4.

DOI: https://doi.org/10.7554/eLife.38060.004

model with the zero voltage C-O equilibrium constant ($L_0$) scaled down from one published value of $1 \times 10^{-6}$ (**Horrigan and Aldrich, 2002**). As shown in **Figures 2C and D**, G–V curves simulated with $L_0$ scaled to 1/3 of the control value are only predicted to be positively shifted by about 20 mV from the control G-V, much less than the gating shift observed in our experiments. The H-A simulation also showed that a 1000-fold reduction in $L_0$ was required to reproduce the observed G-V shift if the only effect of extracellular $H^+$ was to alter the C-O equilibrium of BK channel (**Figure 2C,D**). Therefore, a change in the C-O equilibrium cannot fully account for the gating shift induced by extracellular acidification.

## Gating current measurement shows that activation of BK VSD is strongly inhibited by extracellular $H^+$

To directly determine the effect of extracellular $H^+$ on the function of the BK VSD, we compared the fast components of BK channel gating current ($I_{gfast}$) recorded from giant outside-out patches at $pH_O$ 7 and 4 (**Figure 3A**). The $I_{gfast}$, which reflects the movement of the BK VSD in closed channels, was isolated by fitting the first 100 μs of $I_{gON}$ with an exponential function (**Horrigan and Aldrich, 1999a, 2002**). The fast charge movement associated with VSD activation ($Q_{fast}$) was then determined by integration under the exponential fit. The Boltzmann fit of the $Q_{fast}$-V relationship shows that the $V_h$ of BK VSD activation was positively shifted from 190 mV to 309 mV when $pH_O$ was changed from 7 to 4 (**Figure 3B**). In addition, the slope of $Q_{fast}$-V relationship was reduced from 0.46$e$ to 0.39$e$, indicating that the effective gating charge associated with BK VSD movement was moderately reduced by extracellular acidification. **Figure 3C** shows the G-V curves of BK channels at

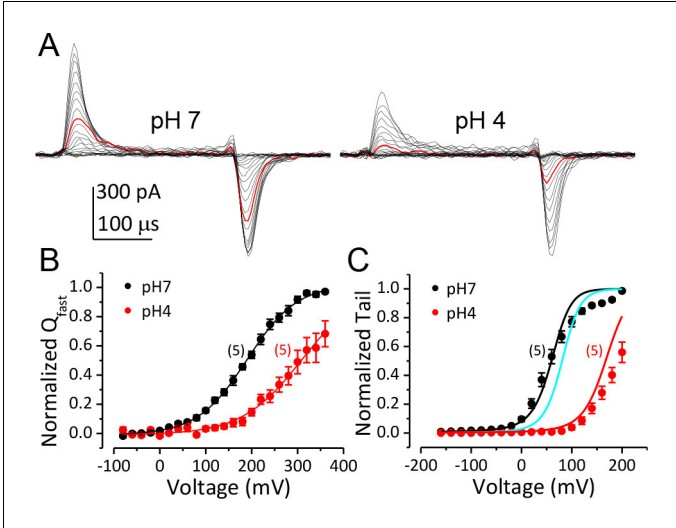

**Figure 3.** Inhibition of BK VSD activation is the primary mechanism for BK gating shifts induced by extracellular H$^+$. (**A**) BK gating currents recorded from a macropatch perfused at pH$_O$ 7 (left) or 4 (right). Test pulses were from −80 to +360 mV in 20 mV increments. Red traces were evoked by 200 mV test pulses. (**B**) Normalized Q$_C$-V relationship of BK at pH$_O$ 7 and 4 averaged from five experiments. Gray lines are Boltzmann fits (pH7: $z_j$ = 0.46 ± 0.02 e, $V_{hC}$ = 190.1 ± 2.7 mV, pH4: $z_j$ = 0.38 ± 0.02 e, $V_{hC}$ = 309.5 ± 2.8 mV). (**C**) BK G-V relationships at pH$_O$ 7 or 4 with 10 μM [Ca$^{2+}$]$_{in}$ (n = 5). The G-V at pH 7 is fit with H-A model using published parameters (**Horrigan and Aldrich, 2002**) as initial values. $z_j$ and $V_{hC}$ are fixed at the values determined from the Q$_C$-V relationship at pH 7. The fit results are: K$_d$ = 3 μM, C = 11, D = 24, E = 1, $L_0$ = 2.7 × 10$^{-6}$, $z_l$ = 0.1 e (black line). The cyan line is a G-V curve calculated using the H-A model with this same set of parameters except that $L_0$ is reduced to 0.9 × 10$^{-6}$. The red line is a G-V calculated using the H-A model with $V_{hC}$ and $z_j$ from the Boltzmann fit of Q$_C$-V at pH$_O$ 4 and reduced $L_0$ (0.9 × 10$^{-6}$).

DOI: https://doi.org/10.7554/eLife.38060.005

pH$_O$ 7 and 4 with 10 μM [Ca$^{2+}$]$_{in}$. We first fit the G-V at pH$_O$ 7 using the H-A model with $V_{hC}$ (half-activating voltage of BK VSD in a closed channel) and $z_j$ (partial charge associated with the BK VSD resting-activated equilibrium constant J) fixed at the values determined from the Q$_{fast}$-V relationship at pH$_O$ 7 and published values (**Horrigan and Aldrich, 2002**) as initial guesses for all other parameters (**Figure 3C**, black line). Then we calculated the predicted BK G-V curves at pH$_O$ 4 based on our measured changes either in $L_0$, in $V_{hC}$ and $z_j$, or both. The calculated GV based on a two-fold reduction of $L_0$ only positively shifted the G-V (**Figure 3C**, cyan line) by less than 20 mV from the G-V at pH 7, while the GV calculated based on changes of BK VSD activation ($V_{hC}$ increased from 190 mV to 309 mV, $z_j$ reduced by 15%) along with the two-fold reduction in $L_0$ reasonably reproduced the gating shift induced by extracellular acidification at pH 4 (**Figure 3C**, red line). Therefore, the primary effect of extracellular acidification on BK channels is to inhibit BK VSD activation.

## Three acidic residues on the extracellular side of BK VSD are involved in BK inhibition by extracellular H$^+$

The IC$_{50}$ for the BK gating shift by extracellular H$^+$ is around pH 4 (**Figure 1C**), suggesting the involvement of acidic residues in BK inhibition by extracellular acidification. There are eight acidic residues on the extracellular side of each BK α-subunit, with four (D133, E139, D147, D153) in the VSD and the other four (E257, E264, E276, D292) in the PGD (**Figure 4A**). A multiple sequence alignment in the VSDs of different members of the voltage-dependent cation channel superfamily shows that the acidic residues in the VSD are largely conserved among these channels, with all four residues conserved between BK and Kv channels. For the four acidic residues in the BK PGD, only the two in the pore-loop (E276 and D292) are conserved between BK and Kv channels (**Figure 4B**). It should be mentioned that D292, a highly conserved residue among most K channels, has been systemically studied in human BK (hSlo1) channels in a previous study, which showed that the G-V of hSlo1 D292N is only moderately shifted toward positive potentials when compared with that of WT

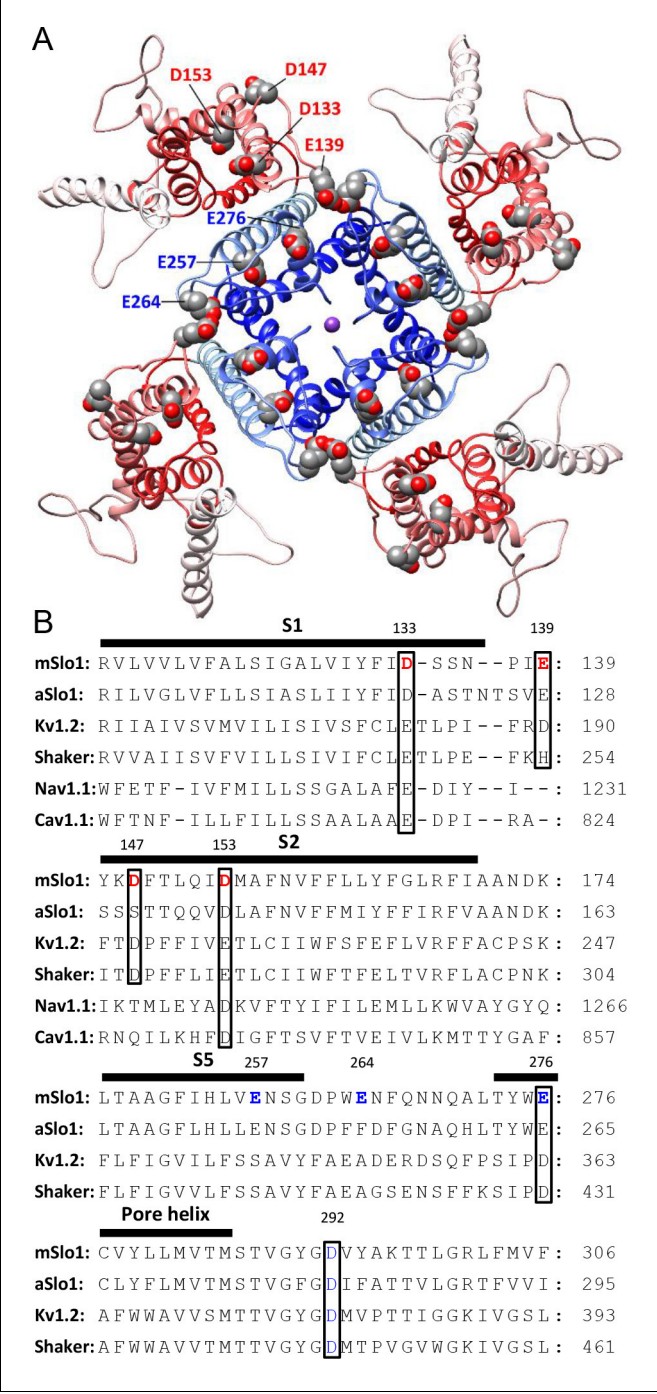

**Figure 4.** BK acidic residues potentially exposed to extracellular medium. (**A**) mSlo1 homology model based on the cryo-EM structure of liganded Aplysia BK channel (PDB: 5tj6) viewed from the extracellular side. Acidic residues potentially exposed to extracellular solution are rendered as spheres with oxygen and carbon colored in red and gray, respectively. The BK VSDs and PGD are colored in red and blue, respectively. The purple dot in the center is $K^+$. (**B**) Multiple sequence alignment of transmembrane segments containing extracellularly accessible acidic residues from mouse and Aplysia homologues of Slo1 (mSlo1, aSlo1), Kv channels (Kv1.2, *Shaker*), and human Nav1.1 and Cav1.1 channels. For the latter two channels, only the DIII VSD segments were included in the alignment. Extracellularly accessible acidic residues conserved among these channels are indicated by boxes. Residues examined in the present study are bold.

DOI: https://doi.org/10.7554/eLife.38060.006

channels (*Haug et al., 2004a*). Therefore, it is unlikely that the large gating shift of BK channels that is induced by extracellular acidification would be mainly caused by reduced side chain charge of D292 at low $pH_O$. In the current study, we individually replaced seven other acidic residues (D133, E139, D147, D153, E257, E264, E276) with alanine and examined sensitivity of the resulting mutants to extracellular $H^+$. Consistent with the result that extracellular $H^+$ positively shifts the gating of BK channels mainly by inhibition of BK VSD activation, all three alanine substitutions that induced a significant reduction in the $pH_O$-dependent gating shift are in the BK VSD (D133A, D147A, D153A).

Among these three key residues, alanine-substitution of D153 had the greatest effect on the $pH_O$-dependent gating shift. As shown in *Figure 5A*, the activation and deactivation kinetics of D153A current at $pH_O$ 4 were not significantly different from that at $pH_O$ 7, even though the amplitude of the current was moderately reduced mostly due to a reduction of single channel conductance at $pH_O$ 4. In accordance with this, the G-V curves of D153A were shifted by less than 10 mV when $pH_O$ was changed from 7 to 4 (*Figure 5B*). Correspondingly, for D153A the change in gating equilibrium free energy by extracellular acidification at $pH_O$ 4 ($\Delta G_0^{pH}=G_0^{pH4}-G_0^{pH7}=z^{pH4}FV_h^{pH4}-z^{pH4}FV_h^{pH7}$) was only 0.3 kcal/mol (*Figure 5I*). As a comparison, the same change of $pH_O$ increased the gating equilibrium free energy of wild type (WT) BK channels by about 2.8 kcal/mol. This result shows that D153 is critical for BK inhibition by extracellular $H^+$.

In addition to D153A, alanine-substitution of D133 or D147 also significantly reduced the gating shifts induced by extracellular acidification, even though neither mutation was as effective as D153A in abolishing the $pH_O$-dependent gating shift. For D133A, changing $pH_O$ from 7 to 4 still permitted a G-V shift of about +50 mV (*Figure 5C and D*), while for D147A, the same extracellular acidification shifted the G-V by more than +60 mV (*Figure 5E and F*). Despite the weaker effects of these mutations relative to D153A, both mutations reduced $\Delta G_0^{pH}$ by more than 1 kcal/mol (*Figure 5I*), indicating that both aspartates do contribute to BK inhibition by extracellular acidification. Double alanine-substitution of D133 and D147 (D133A/D147A) further reduced the magnitude of the $pH_O$-dependent gating shift. The G-V curve of this double mutant was shifted by less than +25 mV when $pH_O$ was changed from 7 to 4 (*Figure 5G and H*). The coupling energy between D133 and D147 can be calculated using double-cycle mutation analysis (*Yifrach and MacKinnon, 2002*) as: $[(\Delta G_0^{pH}{}_{WT}-\Delta G_0^{pH}{}_{D133A})-(\Delta G_0^{pH}{}_{D147A}-\Delta G_0^{pH}{}_{D133AD147A})]$=1.2 kcal/mol, indicating that these two acidic residues may not act independently in producing BK inhibition by extracellular acidification.

By comparing the G-Vs of the above mutants at $pH_O$ 7 with that of WT channels it can be noticed that alanine-substitution at these key sites also induced positive gating shifts. In *Figure 5J* the change of gating equilibrium free energy by alanine-substitution at pH7 ($\Delta G_0^{MUT}=G_0^{MUT}-G_0^{WT}$) is plotted against $\Delta G_0^{pH}$. From this plot it is clear that $\Delta G_0^{MUT}$ and $\Delta G_0^{pH}$ are inversely related, indicating that alanine-substitution of the three key acidic residues and extracellular acidification shift the gating of BK channels by a common mechanism.

## Discussion

In this study we describe a strong and reversible inhibitory effect of extracellular $H^+$ on BK channel activation at $pH_O$ below 6, with particularly profound effects at pH 5 and 4. This inhibition involves multiple mechanisms, with inhibition of BK VSD activation as the primary mechanism. In addition to inhibition of BK VSD, extracellular $H^+$ also induces a moderate change in the C-O equilibrium of channel activation favoring the closed conformation. There is also a direct pore inhibition by extracellular protons, but this effect will be of minor impact in the face of the large gating shifts. The combination of the effects on VSD activation and C-O equilibrium shifts the G-V relationship of BK toward positive potentials by more than 100 mV at $pH_O$ 4. As a result, the activity of BK channels is low in physiological voltage ranges when the channels are exposed to a strongly acidified extracellular environment even with saturating intracellular $Ca^{2+}$. We further show that alanine-substitution of three acidic residues in the extracellular side of BK VSD reduces the gating shift caused by extracellular acidification, suggesting that $H^+$ inhibits BK channel activation mainly by reducing the charge of these acidic residues. Since these three acidic residues are highly conserved among voltage-dependent $K^+$ channels, inhibition of VSD activation could be a universal mechanism for extracellular $H^+$ to regulate the activity of voltage-dependent ion channels that may be exposed to acidic environments.

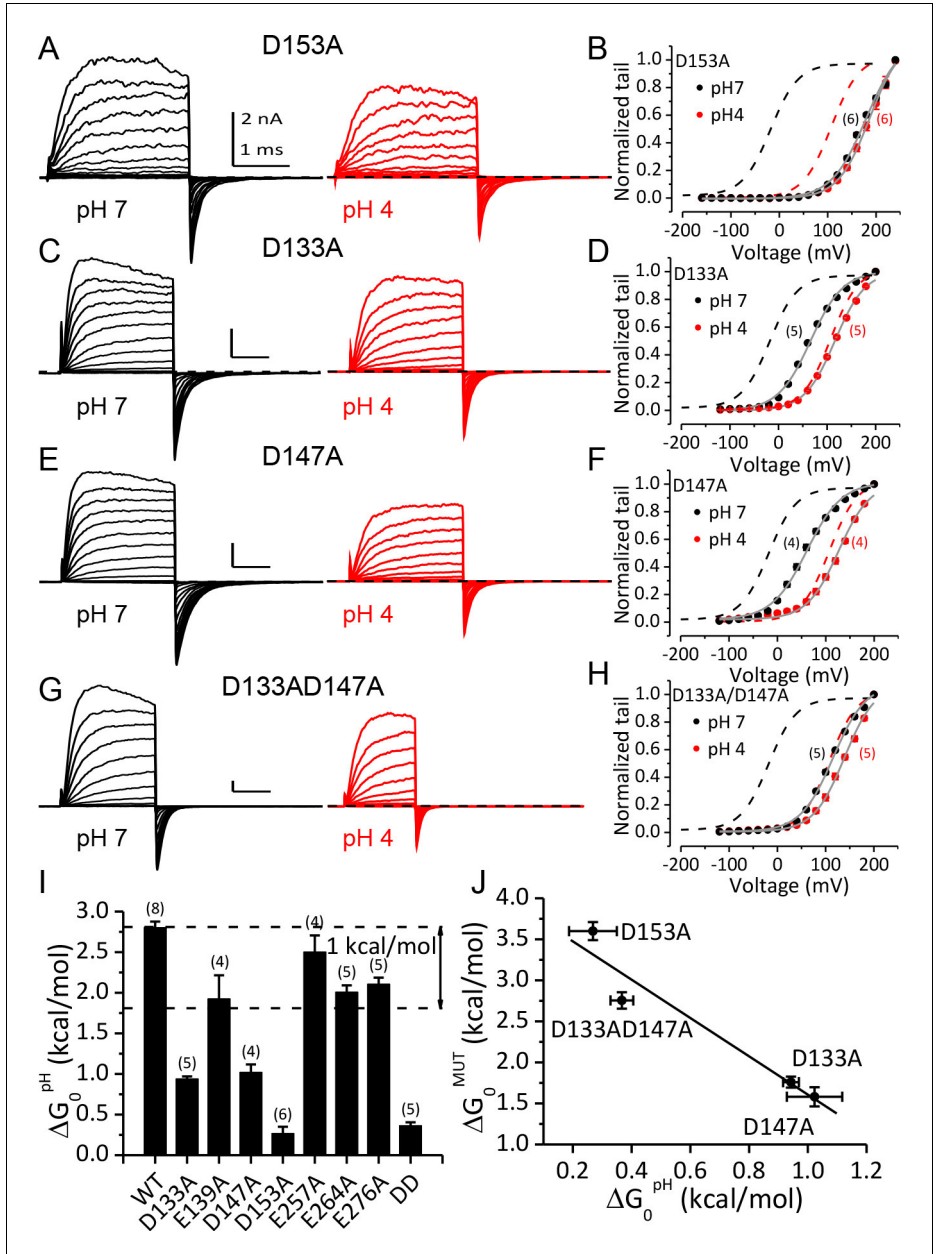

**Figure 5.** Acidic residues involved in BK inhibition induced by extracellular H$^+$. (**A**) Macroscopic currents of mSlo1D153A from an outside-out patch perfused at pH$_O$ 7 (left) or pH$_O$ 4 (right). The currents of D153A were evoked by steps from −160 to +240 mV with 20 mV increments. All other currents shown in this figure were evoked by steps from −120 to +200 mV (20 mV increments). All pipette (intracellular) solutions contained 300 µM Ca$^{2+}$. (**B**) The G-V curves of D153A at pH$_O$ 7 or 4. The number of patches contributing to each set of G-V relationship is given in parentheses in this and following G-V plots. Boltzmann fit results (grey lines) are $z$ = 0.70 ± 0.03 e, $V_h$ = 179.4 ± 2.4 mV (pH 7), $z$ = 0.78 ± 0.03 e, $V_h$ = 186.4 ± 2.2 mV (pH 4), G$_{max}$ = 1.16 ± 0.03. Dotted lines in this and following G-V plots are the G-V curves of WT channels at pH7 (black) or 4 (red) with 300 µM [Ca$^{2+}$]$_{in}$ (**C**) Macroscopic currents of mSlo1D133A at pH$_O$ 7 (left) or pH 4 (right). (**D**) The G-V curves of D133A at pH$_O$ 7 or 4. Boltzmann fit results (grey lines) are $z$ = 0.76 ± 0.03 e, $V_h$ = 66.1 ± 1.7 mV (pH 7), $z$ = 0.81 ± 0.03 e, $V_h$ = 116.2 ± 1.6 mV (pH 4), G$_{max}$ = 1.00 ± 0.02. (**E**) Macroscopic currents of mSlo1D147A at pH$_O$ 7 (left) or pH 4 (right). (**F**) The G-V curves of D147A at pH$_O$ 7 or 4. Boltzmann fit results (grey lines) are $z$ = 0.70 ± 0.04 $e$, $V_h$ = 60.8 ± 2.6 mV (pH 7), $z$ = 0.77 ± 0.05 e, $V_h$ = 126.7 ± 2.3 mV (pH 4), G$_{max}$ = 0.99 ± 0.02. (**G**) Macroscopic currents of mSlo1D133A147A at pH$_O$ 7 (left) or pH 4 (right). (**H**) The G-V curves of mSlo1D133AD147A at pH$_O$ 7 or 4. Boltzmann fit results (grey lines) are $z$ = 0.74 ± 0.04 e, $V_h$ = 115.6 ± 2.3 mV (pH 7), $z$ = 0.78 ± 0.04 e, $V_h$ = 138.9 ± 2.1 mV (pH 4), G$_{max}$ = 1.07 ± 0.03. (**I**) Change of gating equilibrium free energy by extracellular

*Figure 5 continued on next page*

*Figure 5 continued*

acidification($\Delta G_0^{pH}$) calculated from the Boltzmann fits of G-V curves. The number of experiments for each construct is listed above each column. (J) The change of gating equilibrium free energy by alanine substitution ($\Delta G_0^{MUT}$) plotted against the change of gating equilibrium free energy by extracellular acidification ($\Delta G_0^{pH}$). The solid line is a linear fit with R of 0.96. Standard error in $\Delta G_0^{MUT}$ ($\alpha\Delta_{G0MUT}$) is calculated according to: $\alpha\Delta_{G0MUT} = (\alpha^2_{G0WT} - \alpha^2_{G0MUT})^{1/2}$.

DOI: https://doi.org/10.7554/eLife.38060.007

## Relation to previous studies and structural implications

Some previous studies have examined aspects of the effect of extracellular acidification on Kv channels, usually at more moderate pHs above 5.0 (*Deutsch and Lee, 1989*; *Horrigan and Aldrich, 1999a*; *Claydon et al., 2000*; *Prole et al., 2003*; *Kwan et al., 2006*; *Claydon et al., 2007*, *Claydon et al., 2008*). At such concentrations, extracellular H[+] has been observed to act primarily in the pore region of these channels to reduce current amplitude. For example, $pH_O$ 5.9 inhibits Kv 1.5 channels by accelerating entry of open channels into P/C-type inactivation and by stabilizing closed channels in closed-inactivated states (*Claydon et al., 2008*). In the current study, we show that strong acidification at pHs lower than five induces additional inhibition in the BK VSD, most likely by reducing the side chain charges of three key acidic residues on the extracellular side of BK VSD. Given that these three residues are highly conserved between BK and Kv channels, it is reasonable to assume that the activation of Kv VSD could also be inhibited by such strong extracellular acidification. Indeed, it has been shown that the G-V and Q-V relationships of Kv1.5 were also significantly shifted toward positive potentials at $pH_O$ 5 or lower (*Claydon et al., 2007*).

Among the three key residues involved in BK inhibition by extracellular H[+], D153 in the S2 of BK VSD appears to be the most important one as D153A almost completely abolishes the BK gating shift induced by extracellular acidification. D153 was identified as a potential voltage-sensing residue for BK channels by a previous mutagenesis analysis of the BK VSD (*Ma et al., 2006*). In this work the authors analyzed BK ionic current within the framework of the H-A model to show that substitution of D153 by cysteine reduces the $z_j$ of BK channels by 40%. In agreement with this result, by examining gating currents recorded at different $pH_O$s, we found that the $z_j$ of BK channels was reduced by extracellular H[+] (*Figure 3B*). It should be noted that the $z_j$ of BK channels was only reduced by 15% at $pH_O$ 4, suggesting that the side chain of D153 may still be partially charged at $pH_O$ 4. In accordance with this, the BK gating shift induced by changing $pH_O$ from 7 to 4 was less than that induced by the D153A mutation (*Figure 5B*).

To understand the role of D153 in BK VSD activation, we generated a pair of homologous mSlo1 structures using two recently published Aplysia BK (aBK) channel cryo-EM structures as templates (*Tao et al., 2017*; *Hite et al., 2017*). The one based on the structure solved in the presence of saturating intracellular $Ca^{2+}$ and $Mg^{2+}$ is termed 'liganded' and the other based on the structure solved in the absence of intracellular divalent cation is termed 'metal-free'. *Figure 6A* shows the VSDs of these two mSlo1 structures superimposed by their pore-loops. It can be seen that D153 is located at the bottom of a crevice on the extracellular end of the BK VSD, with little displacement (<1 Å) of D153 between these two structures. As a comparison, the side chain of R213, another primary voltage sensing residue located in the BK S4 (*Ma et al., 2006*) is shifted toward the extracellular side by more than 3.5 Å in the liganded structure. Based on the functional importance of the free carboxyls of D153 in gating, we searched for potential hydrogen bonds that might involve the side chain of D153 in the two mSlo1 structures. The result shows that D153 may form candidate hydrogen bonds with several different S4 residues in the two structures. In the metal-free structure, one of the carboxyl oxygens in the side chain of D153 is within the hydrogen-bonding distance with the δ-nitrogen of R207 (3.1 Å, *Figure 6B*), while in the liganded structure both carboxyl oxygens of D153 are aligned with the two ω-nitrogens of R210 for potentially strong hydrogen bonding (*Figure 6C*). These observations are in accordance with the result from a previous study of BK inhibition by extracellular $Cu^{2+}$, which shows that both D153 and R207 are required for $Cu^{2+}$ coordination in the BK VSD in resting states (*Ma et al., 2008*). Thus, it is conceivable that D153 might interact with various basic residues in BK S4 to stabilize the VSD in particular transitional conformations on the pathway to activation. If this was true, then extracellular acidification could inhibit BK VSD activation by

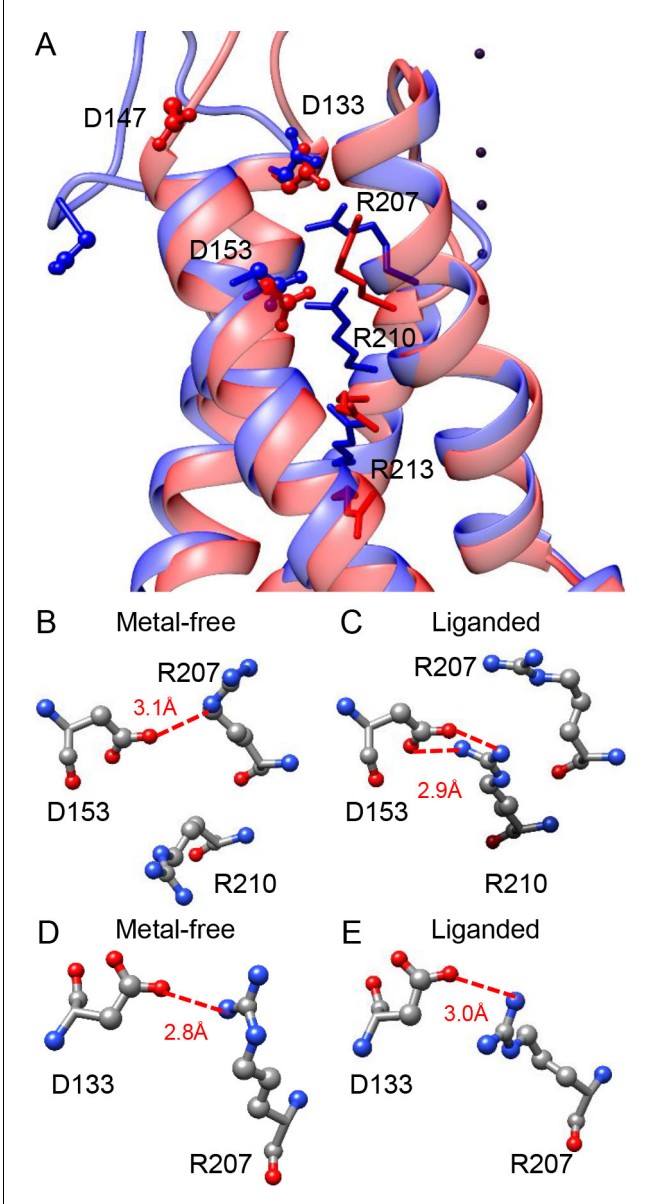

**Figure 6.** Potential interactions involving the three key residues in the BK VSD. (**A**) The VSDs of metal-free (red) and liganded (blue) mSlo1 structures with three key acidic residues rendered as ball-and-chain and three conserved S4 arginines (homologous to Shaker R2, R3 and R4) rendered as sticks. The four dark purple dots are K⁺s in the BK selectivity filter, which are included as a reference for vertical displacement. The two structures are superimposed by the pore-loop. (**B**) D153 may form a hydrogen bond with R207 in the metal-free mSlo1 structure. The distance between the side chains of D153 and R210 is more than 6 Å, which is too far for hydrogen bonding. (**C**) D153 may form a hydrogen bond with R210 in the liganded mSlo1 structure. The distance between the side chains of D153 and R207 is more than 4 Å, which is also beyond the limit of effective hydrogen bonding. (**D–E**) The distances between the side chains of D133 and R207 in both the metal-free (**D**) and the liganded (**E**) structures are within 3 Å, allowing potential interaction between D133 and R207 in both conformations.
DOI: https://doi.org/10.7554/eLife.38060.008

weakening such interactions. Indeed, previous work shows that residues with opposite charges placed at *Shaker* 283 and *Shaker* 368 (homologous to mSlo1 R210) can stabilize the VSD of the resulting *Shaker* channel in a partially active conformation (*Tiwari-Woodruff et al., 2000*).

In addition to D153, we also identified two other acidic residues (D133 and D147) that are important for BK inhibition by extracellular $H^+$ and therefore important for the function of the BK VSD. Replacing each of these two asparates with alanine causes a significant reduction in the $pH_O$-dependent gating shift, while the impact of the double alanine-substitution (D133AD147A) on the gating shift induced by extracellular acidification is comparable to that of D153A. D133 and D147 are more exposed to the extracellular environment than D153 in both mSlo1 structures (*Figure 6A*). For D133 in the S1 helix of BK VSD, there is little change in position between the two structures. The side chain of D133 is close to R207 in both mSlo1 structures (*Figure 6D–E*), indicating that D133 may also form important interactions with S4 residues to stabilize BK VSD. Interestingly, D133 is also involved in $Cu^{2+}$ coordination with the BK VSD (*Ma et al., 2008*). The lack of change in distance between D133 and R207 in the two mSlo1 structures indicates that either the state-dependence of $Cu^{2+}$-VSD coordination is largely determined by the distance between D153 and R207, or the conformation captured by the liganded structure represents a partially activated VSD state. For D147, a large displacement is observed between the two structures, which is not totally unexpected as D147 is in the flexible S1-S2 linker. However, this displacement is largely horizontal. Thus, D147 remains on the surface of the BK VSD in both structures. Unlike that of D153 and D133, the side chain of D147 is not close to any residue with potential for strong interaction in the two mSlo1 structure, indicating that D147 may not contribute to VSD activation by interacting with other residues to stabilize transitional conformations during VSD activation. Instead, D147 may participate in BK VSD activation by altering local electric field.

The sequence alignment in *Figure 4B* shows that D133 and D153 are highly conserved among the members of the voltage-dependent cation channel superfamily. Previous studies (*Long et al., 2007*; *Pathak et al., 2007*; *Tao et al., 2010*; *Hoshi and Armstrong, 2012*) suggest that these acidic residues may form critical interactions with S4 basic residues during the activation of Kv channels. For example, it has been reported that extracellularly applied $La^{3+}$ interacts with Kv1.2 E183 and E226 (homologous to mSlo1 D133 and D153, respectively) to inhibit the activation of Kv VSD (*Hoshi and Armstrong, 2012*), a mechanism that may underlie the inhibitory effect of extracellular divalent cations observed in Kv (*Gilly and Armstrong, 1982*; *Zhang et al., 2001a*) and Nav (*Frankenhaeuser and Hodgkin, 1957*; *Hille et al., 1975*) channels. Therefore, it is reasonable to assume that extracellular $H^+$ may also inhibit the activation of members in the voltage-dependent cation channel superfamily by a similar mechanism.

In the current study we found that the D153A mutation alone can almost completely abolish the gating shift induced by extracellular acidification. However, the shallow slope of the $pH_O$-dependent gating-shift curve shown in *Figure 1E* indicates that multiple sites with different pKa values may be involved in BK inhibition by extracellular $H^+$. Previous studies show that, depending on aqueous exposure and interaction with nearby charged residues, the pKa of Asp and Glu can differ substantially from the expected value of around 4 (*Forsyth et al., 2002*; *Ma et al., 2008*). Therefore, it is possible that the three acidic residues identified in this study may be exposed to different local environments that result in somewhat modified pKa values. It is also possible that another extracellular residue with higher pKa is involved in BK regulation by extracellular $H^+$. One possible candidate for an additional $pH_O$ sensor is mSlo1 H254 in the S5 of BK PGD. These intriguing possibilities provide strong motivation for further investigation of BK regulation by extracellular $H^+$.

## Physiological implications

As an ion channel with great physiological importance, the BK channel is a popular subject for research on ion channel regulation. Whereas regulation of BK channels by cytosolic factors has been extensively studied (*Hou et al., 2009*), much less is known about how physiological signals may regulate BK channels on the extracellular face of the BK channel. Two recent studies suggest that BK channels are ubiquitously expressed in lysosomal membranes in both excitable and non-excitable cells (*Cao et al., 2015*; *Wang et al., 2017*). The results from these studies suggest that lysosomal BK channels are functionally coupled with the lysosomal $Ca^{2+}$-permeable channel TRPML1 to regulate lysosomal $Ca^{2+}$ release/refilling, which is a key process underlying lysosomal physiology (*Xu and Ren, 2015*). Since the extracellular side of the lysosomal BK channel is expected to be exposed to the acidic environment of the lysosome lumen, the basal activity of the lysosomal BK channel is expected to be low. However, it should be noted that lysosomal pH is highly heterogeneous, with the luminal pH of some primary lysosomes close to 7 (*Johnson et al., 2016*; *Bright et al., 2016*).

Therefore, it is conceivable that the activity of lysosomal BK channels might be regulated by lyso-somal luminal pH. Interestingly, BK channels are also sensitive to intracellular $H^+$, which mimics the effect of $Ca^{2+}$-binding in the BK RCK1 domain (*Avdonin et al., 2003*; *Hou et al., 2008a*). Thus, intra-cellular $H^+$ may enhance the activity of BK channels in the absence of intracellular $Ca^{2+}$, while during elevations of intracellular $Ca^{2+}$, intracellular $H^+$ may inhibit BK channel activity (*Cook et al., 1984*; *Christensen and Zeuthen, 1987*; *Schubert et al., 2001*; *Guarina et al., 2017*), presumably by inter-fering with $Ca^{2+}$-binding. In any case, it is clear that the present results require that any consider-ation of the role of BK channels in lysosomes must take into account its regulation by luminal pH.

In excitable cells, might changes in extracellular pH affect excitability? Although it seems unlikely that neurons will be exposed to pH as low as those required to inhibit BK channels in the presence study, a recent study has observed that BK channels in mouse chromaffin cells (MCC) are inhibited by extracellular $H^+$ (*Guarina et al., 2017*). The mechanism underlying this inhibition appears to be different from that described in the current study for two reasons: first, the $IC_{50}$ of MCC BK inhibi-tion by extracellular $H^+$ is 0.1 μM (pH 7), which is lower than what we observed in our experiment (52 μM/pH 4.3). Second, the onset (>30 s) and recovery (~3 min) of MCC BK inhibition by extracellu-lar $H^+$ is much slower than that of the inhibition described here (<1 s for both onset and recovery). Based on the $IC_{50}$ and the time course of MCC BK inhibition by extracellular $H^+$, it seems likely that at least some of the BK inhibition observed in CCs with extracellular acidification may reflect the time course of intracellular acidification induced from elevation of extracellular $[H^+]$.

## Materials and methods

### General methods

Oocyte preparation, handling of RNA, and electrophysiological methods used here were identical to those described in other papers from this laboratory (*Zhang et al., 2001a*; *Zhou et al., 2010*; *Zhou and Lingle, 2014*). In brief, the mouse homologue of BK channels (mSlo1, mKCa1.1) were expressed in *Xenopus* oocytes by RNA injection. All BK ionic and gating currents presented in the current work were recorded from excised patches in the outside-out configuration (*Hamill et al., 1981*). The pipette resistance for ionic current recording was typically 1–2 MΩ after heat polishing. The pipette solution (bathing the cytoplasmic face of patch membranes) contained (in mM): 140 K-methanesulfonate (K-MES), 20 KOH, 10 HEPES, with pH adjusted to 7.0 by methanesulfonic acid.5 mM EGTA was included in 0 $Ca^{2+}$ pipette solution. For 10 μM $Ca^{2+}$ pipette solution, 5 mM HEDTA was used as $Ca^{2+}$ buffer. For 300 μM $Ca^{2+}$ pipette solution, no $Ca^{2+}$ buffer was added. 10 μM $Ca^{2+}$ solution was prepared by adding 1M Ca-MES stock solution to obtain the desired $[Ca^{2+}]$, as defined with a $Ca^{2+}$-sensitive electrode calibrated with commercial $Ca^{2+}$ solutions (WPI, Sarasota, FL). Giga-ohm seals were formed while the oocytes were bathed in frog Ringer (in mM, 115 NaCl, 2.5 KCl, 1.8 $CaCl_2$, 10 HEPES, pH 7.4). Following patch excision, the pipette tip was moved into flowing test sol-utions. Solution exchange at the tip of recording pipette was accomplished with an SF-77B fast per-fusion stepper system (Warner Instruments, Hamden, CT). The composition of the solution used to bath the extracellular membrane face was (in mM) 140 KMES, 20 KOH, 2 $MgCl_2$, 10 HEPES, with pH adjusted to desired concentration by methanesulfonic acid.

Gating current was recorded from outside-out macropatches using pipettes with tip diameters of 10–20 μm. The internal pipette solution contained (mM): 141 NMDG, 135 methanesulfonic acid, 10 HEPES, and 5 EGTA, pH 7.0. The external solution contained (mM): 127 tetraethylammonium hydroxide, 125 methanesulfonic acid, 2 $MgCl_2$, and 10 HEPES, with pH adjusted to desired value using methanesulfonic acid. Both voltage command and current output were filtered at 20 kHz with 8-pole Bessel filters (Frequency Devices) to avoid saturation by fast capacitive transients (*Horrigan and Aldrich, 1999a*). Gating currents were sampled at 100 kHz. A P/−6 protocol was used for leak subtraction from a holding potential of −120 mV (*Armstrong and Bezanilla, 1974*).

All experiments were performed at room temperature (~22–25°C). pH of extracellular solutions was checked daily before recording. All chemicals were purchased from Sigma-Aldrich.

### Data analysis

The G-V relationship of BK channels was determined from tail currents measured 150 μs after repo-larization to −120 mV. G-V curves were fit by a Boltzmann function of the form

of $G(V) = \frac{G_{max}}{1+exp(-z(V-V_h)/kt)}$, where $G_{max}$ is maximal conductance, $z$ is apparent voltage-dependence in units of elementary charge, $V_h$ is the voltage of half-maximal activation, and $k$ and T have their usual physical meanings. The Horrigan-Aldrich (H-A) model (*Horrigan and Aldrich, 2002*) was also used for G-V curve fitting and simulation. This model describes the gating of BK channel as three sets of allosterically coupled conformational changes reflecting the three functional modules of a BK channel: (1) the closed-open equilibrium in the PGD (C-O, equilibrium constant $L = L_0exp(z_LV/kT)$, where $L_0$ and $z_L$ are the zero voltage value of $L$ and its partial charge, respectively), (2) resting-activated equilibrium in the VSD (R-A, equilibrium constant $J = J_0exp(z_JV/kT)=exp(z_J(V-V_{hC})/kT)$, where $J_0$ and $z_J$ are the zero voltage value of $J$ and its partial charge, respectively, while $V_{hC}$ is the half-activating voltage of BK VSD in a closed channel), and (3) $Ca^{2+}$ binding in the CTD (X-X•$Ca^{2+}$, binding constant $K$). The coupling between PGD and VSD is represented by an allosteric factor D such that $L$ increases D-fold with the activation of each VSD. Similarly, interactions of $Ca^{2+}$-binding with PGD and VSD are represented by allosteric factors C and E, respectively. This model defines the steady-state open probability ($P_O$) of BK channel as

$$P_O = \frac{L(1+KC+JD+JKCDE)^4}{L(1+KC+JD+JKCDE)^4+(1+K+J+JKE)^4}$$

The voltage-dependent change of BK single channel current induced by extracellular acidification was fit by the Woodhull model (*Woodhull, 1973*): $\frac{i_{pH4}}{i_{pH7}} = \frac{k_{d0}+10^{-7}exp(-\delta zVF/RT)}{k_d0+10^{-4}exp(-\delta zVF/RT)}$, in which $i_{pH4}$ and $i_{pH7}$ are the unitary current at $pH_O$ 4 and $pH_O$ 7, respectively, $k_{d0}$ is the dissociation constant of blocking ion at 0 mV, V is the transmembrane potential, $\delta$ is the fraction of the membrane potential at the blocking ion binding site, and $z$ is the valence of the blocking ion ($z$ = 1 e for $H^+$). $F$, $R$ and $T$ have their usual physical meanings.

Open probability times the number of BK channels ($nP_O$) at negative potentials (limiting $nP_O$) was determined from single channel traces using the threshold-based Event Detection mode in Clampfit 9.2 (Molecular Device, RRID:SCR_011323). Homologous structures of mSlo1 were built in MODEL-LER 9 (*Sali and Blundell, 1993*, RRID:SCR_008395) using the Cryo-EM structures of the Aplysia BK channel as templates (*Tao et al., 2017*; *Hite et al., 2017*). Structure analysis and structure image preparation were accomplished using UCSF Chimera (*Pettersen et al., 2004*, RRID:SCR_004097). Data were analyzed using OriginPro 7.5 (OriginLab Corporation, RRID:SCR_015636) or programs developed in this laboratory. Error bars in the figures represent SEMs. Some error bars are smaller than symbol size. Curve fitting results are reported as the fitted values with standard error, which was estimated by OriginPro according to the Error Propagation formula.

## Acknowledgements

This work was supported by NIH grant GM118114 to CJL.

## Additional information

### Funding

| Funder | Grant reference number | Author |
| --- | --- | --- |
| National Institute of General Medical Sciences | GM118114 | Christopher J Lingle |

The funders had no role in study design, data collection and interpretation, or the decision to submit the work for publication.

### Author contributions

Yu Zhou, Conceptualization, Data curation, Formal analysis, Investigation, Methodology, Writing—original draft, Writing—review and editing; Xiao-Ming Xia, Investigation, Methodology, Writing—review and editing; Christopher J Lingle, Conceptualization, Funding acquisition, Writing—original draft, Project administration, Writing—review and editing

## Author ORCIDs

Yu Zhou http://orcid.org/0000-0002-1623-7948

## Ethics

Animal experimentation: This study was performed in strict accordance with the recommendations in the Guide for the Care and Use of Laboratory Animals of the National Institutes of Health. All of the animals were handled according to approved institutional animal care and use committee (IACUC) protocols (#20140150) of the Washington University School of Medicine. All surgery was performed under tricaine anesthesia, and every effort was made to minimize suffering.

## Decision letter and Author response

Decision letter https://doi.org/10.7554/eLife.38060.011
Author response https://doi.org/10.7554/eLife.38060.012

# Additional files

## Supplementary files

• Transparent reporting form
DOI: https://doi.org/10.7554/eLife.38060.009

## Data availability

All data generated or analysed during this study are included in the manuscript and supporting files.

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
