## [Decision Letter]

Thank you for submitting your article "BK channel inhibition by strong extracellular acidification" for consideration by *eLife*. Your article has been reviewed by three peer reviewers, and the evaluation has been overseen by Richard Aldrich as the Senior and Reviewing Editor. The following individual involved in review of your submission has agreed to reveal his identity: Frank Horrigan (Reviewer #3).

The reviewers have discussed the reviews with one another and the Reviewing Editor has drafted this decision to help you prepare a revised submission.

Summary:

This manuscript describes an inhibitory effect of low extracellular pH on BK channels, which may be relevant to their physiological function in lysosomes or in other tissues under pathophysiological conditions. The mechanism of inhibition is studied in detail using ionic and gating currents, site-directed mutagenesis, and H-A model analysis to reveal that low pH primarily inhibits voltage-sensor activation and charge movement by acting on three aspartic acidic residues in the voltage-sensor domain.

Essential revisions:

Because the necessary and suggested revisions are fairly minor and straightforward they are included here in the original reviews.

Reviewer #1:

Zhou et al. provide straightforward explanations of why very low pH on the (normally) "extracellular" side inhibits the Slo1 BK channel activity. The overall experimental design, data collection, and data interpretations are sound, and the results and the conclusions are most probably firm.

The conclusion that the interactions of H^+^ with select aspartic acid residues in S1/S2 of Slo1 contribute most to the inhibitory action of "extracellular" H^+^ seems solid. There are two areas that the manuscript/study need to be revised.

First, throughout the study, the authors need to specify how many observations contributed to the results presented. When appropriate, the authors need to specify what each error/variability number represents (e.g., SD, SEM, confidence interval).

Second, the authors should incorporate the VSD activation mechanism information from the previous work on Kv channels. The sequence alignment indicates mSlo1 D133, D147, and D153 are equivalent to E183, E220, and D226 in Kv1.2/2.1 of Long et al. (2017). The activated and (hypothetical) closed conformations of the Kv1.2/2.1 VSD in Long et al. (2017) are quite similar to those presented in Figure 6A. Some discussions about the conserved VSD mechanisms are appropriate. Further, Hoshi and Armstrong (PDBID 22802655) suggested, based on gating and ionic current measurements, that some divalent and trivalent cations may interact with E226 in Kv1.2/2.1 (mSlo1 D153) (albeit the GV shift is considerably smaller than that presented here). Similar mechanisms may be involved in the effect of divalent cations in Kv channels and H^+^ in Slo1.

• Introduction, first paragraph. Perhaps, insert "potentially" into "[…] or even in various organelles"

• "Extracellular". Somehow qualify that the normally extracellular side is the luminal side early on.

• Figure 1D. pH 4 data. Instead of scaling to the observed maximal value, it might be better to scale to the Boltzmann-fit estimated maximal value? In the current form, the max fit value is >1 and the graph probably underestimates the V_h_ shift.

• Figure 2C. It is better not to connect the data points with straight lines.

• Results, section ‘BK channel activation is strongly inhibited at pHO lower than 5’: "[…] not the reduction in single-channel conductance." If illustrative opening records are available, the authors should show them.

• Results, section ‘A change in the C-O equilibrium only accounts for a small portion of the gating shift induced by extracellular H^+^’: "about 2-fold with" and " reduced by 2-fold are" The results seem to show that the single-channel nP_o_ decreased by 3-fold?

*Reviewer #2:*

The manuscript from Zhou et al. reports on effects of extracellular acidification on BK channel gating. Increasing external [H^+^] over the range of 1e-9 to 1e-4 M is observed to slightly decrease single channel conductance and produce a large shift in Vhalf (V_h_). The gating shift is attributed primarily to titration of the side chain of D153 (and to a lesser extent D133 and D147). D153 in particular can form apparent salt-bridge interactions with R207 or R210 in the BK channel VSD that depend on the metal-liganded state of the channel.

The manuscript is in general well-written and the authors working hypothesis is supported by their experimental data, which are of high quality, as well as the available BK channel structural data. The physiological importance of BK channels in many tissues, including loci such as kidney epithelium and intracellular membrane where external acidification may actually modulate these channels, adds to the significance and level of interest to a broad audience. I have only some concerns with the presentation of some of the data, plus a more moderate concern on the experiments regarding C-O equilibrium.

In Figure 1C: It's not clear how the single-channel i-V data, which is on an absolute scale, is related to the model-generated plot of fractional current. Plotting the data in this way seems to suggest that if the model were correct, then the model-generated curve should superimpose on the data, but this is not the case. One possible remedy would be to plot the experimental data as fractional current to compare it to the curve.

For measurements aimed at quantifying the effect of pHo on C-O equilibrium:

1) In the interest of transparency, the authors should list the actual parameter values used to generate the simulated curves in Figure 2B (there are three sets of parameters listed in Horrigan and Aldrich 2002).

2) It is a moderate concern that the estimates of C-O equilibrium are from channel activity with 10 µM Ca at the cytosolic side of the patch, at -100 mV. Under these conditions it is very unlikely that the C-O step has been isolated from effects of voltage sensor movement. These experiments should ideally be performed in that negative Vm range with nominally 0 Ca at the cytosolic side of the patch (also, currents should be measured over a range of negative voltages to confirm that voltage sensor activation is not contributing to gating).

Introduction, "[…] whether BK channels is sensitive" should read "whether BK channels are sensitive"

Also "Given the possible presence of BK channels […]" should probably read "Given the presence of BK channels", unless the authors have reason to doubt their presence at these loci. Same for "Given the possibility that BK channels may be expressed".

Reviewer #3:

This manuscript describes an inhibitory effect of low extracellular pH on BK channels, which may be relevant to their physiological function in lysosomes or in other tissues under pathophysiological conditions. The mechanism of inhibition is studied in detail using ionic and gating currents, site-directed mutagenesis, and H-A model analysis to reveal that low pH primarily inhibits voltage-sensor activation and charge movement by acting on three aspartic acidic residues in the voltage-sensor domain. The manuscript is well written. The data are of high quality, clearly described, and convincing. I have no major concerns.

I do, however, have some questions and a suggestion for a point of discussion. While the apparent IC_50_ of inhibition at pH 3.8 does implicate acidic residues, the pH-dependence is very shallow with a hill coefficient of 0.41, such that shifts in activation are observed even at pH 7 or 6 (relative to pH 9). The mechanism underlying this shallow pH-dependence is not addressed, but is interesting because it makes the effect potentially relevant to physiological or pathophysiological pH conditions other than the extreme example of lysosomes. The shallow pH-dependence would seem to suggest either that (1) the 3 aspartic acids identified may have very different pKa's or (2) or a high pKa sensor was missed. Therefore, some relevant questions are: Do acidic residue mutations reduce the response to pH 6 as well as pH 4 (suggesting they account for the effects near physiological pH)? and, was the effect of mutating the only external histidine in hSlo (H254) tested?

Even if none of these experiments were done, I think it may be worth discussing that the broad pH-dependence could arise from D133, D147, D153 having different pKa, since this fits into the existing discussion. It is well known that the pKa of acidic residues can be altered depending on aqueous exposure and interaction with other charged residues, which the authors show are different for D133, D147, D153, and may change with voltage-sensor activation. In addition, there is previous evidence to suggest that D133 and/or D153 might have an unusually high pKa (at least in the resting state) because the effect of Cu^2+^ reported by Ma et al. 2008 and referenced in the discussion, was dependent on D133 and D153, but not H254, and yet exhibited a relatively high apparent pKa of 6.0.

---

## [Author Response]

Reviewer #1:[…] There are two areas that the manuscript/study need to be revised.First, throughout the study, the authors need to specify how many observations contributed to the results presented. When appropriate, the authors need to specify what each error/variability number represents (e.g., SD, SEM, confidence interval).

The number of experiments for each data point is added to corresponding plots. Information about variability of measurement and uncertainty of curve fitting is added to the Materials and methods subsection “Data analysis”.

Second, the authors should incorporate the VSD activation mechanism information from the previous work on Kv channels. The sequence alignment indicates mSlo1 D133, D147, and D153 are equivalent to E183, E220, and D226 in Kv1.2/2.1 of Long et al. (2017). The activated and (hypothetical) closed conformations of the Kv1.2/2.1 VSD in Long et al. (2017) are quite similar to those presented in Figure 6A. Some discussions about the conserved VSD mechanisms are appropriate. Further, Hoshi and Armstrong (PDBID 22802655) suggested, based on gating and ionic current measurements, that some divalent and trivalent cations may interact with E226 in Kv1.2/2.1 (mSlo1 D153) (albeit the GV shift is considerably smaller than that presented here). Similar mechanisms may be involved in the effect of divalent cations in Kv channels and H^+^ in Slo1.

We added a new paragraph in the subsection “Relation to previous studies and structural implications” to discuss the relationship between previous studies on the VSD activation mechanism and our work on pH inhibition of BK channels based on this comment.

• Introduction, first paragraph. Perhaps, insert "potentially" into "[…] or even in various organelles”

Done.

• "Extracellular". Somehow qualify that the normally extracellular side is the luminal side early on.

We added several sentences in the Introduction to make it clear that the normally extracellular side is the luminal side in some organelles.

• Figure 1D. pH 4 data. Instead of scaling to the observed maximal value, it might be better to scale to the Boltzmann-fit estimated maximal value? In the current form, the max fit value is >1 and the graph probably underestimates the V_h_ shift.

We fit the data in Figure 1D with G_max_ allowed to vary at each pH_O_ and updated Figure 1D and its legend with the new fitting result. The new estimations of V_h_ and *z* are almost identical to the original values.

• Figure 2C. It is better not to connect the data points with straight lines.

The data points are now connected with cubic spline line.

• Results, section ‘BK channel activation is strongly inhibited at pHO lower than 5’: "[…] not the reduction in single-channel conductance." If illustrative opening records are available, the authors should show them.

Representative single channel recordings at pH_O_ 7 and 4 are included in Figure 1—figure supplement 1.

• Results, section ‘A change in the C-O equilibrium only accounts for a small portion of the gating shift induced by extracellular H^+^’: "about 2-fold with" and " reduced by 2-fold are" The results seem to show that the single-channel nP_o_ decreased by 3-fold?

Now these two sentences are changed to “*L* of BK channels at pH_O_ 4 was about 1/3 of that at pH_O_ 7” and “with L_0_ scaled to 1/3 of the control value” respectively.

Reviewer #2:[…] I have only some concerns with the presentation of some of the data, plus a more moderate concern on the experiments regarding C-O equilibrium.In Figure 1C: It's not clear how the single-channel i-V data, which is on an absolute scale, is related to the model-generated plot of fractional current. Plotting the data in this way seems to suggest that if the model were correct, then the model-generated curve should superimpose on the data, but this is not the case. One possible remedy would be to plot the experimental data as fractional current to compare it to the curve.

We put a fractional single channel current-voltage plot and corresponding Woodhull model fit in Figure 1C and provided sample traces of BK single channel current and i-V plot in Figure. 1—figure supplement 1.

For measurements aimed at quantifying the effect of pHo on C-O equilibrium:1) In the interest of transparency, the authors should list the actual parameter values used to generate the simulated curves in Figure 2B (there are three sets of parameters listed in Horrigan and Aldrich 2002).

The parameters used for simulation are added to Figure 2 legend.

2) It is a moderate concern that the estimates of C-O equilibrium are from channel activity with 10 µM Ca at the cytosolic side of the patch, at -100 mV. Under these conditions it is very unlikely that the C-O step has been isolated from effects of voltage sensor movement. These experiments should ideally be performed in that negative Vm range with nominally 0 Ca at the cytosolic side of the patch (also, currents should be measured over a range of negative voltages to confirm that voltage sensor activation is not contributing to gating).

We attempted to measure BK limiting nP_O_ at pH_O_ 4 with 0 [Ca^2+^]_in_. But the openings under such conditions were too brief for us reliably determine nP_O_. Two previous studies (Horrigan and Aldrich, 2002; Carrasquel-Ursulaez et al., 2015) show that there is little BK VSD movement at voltages negative to -100 mV even with 70 to 100 μm [Ca^2+^]_in_. Therefore, BK VSDs should largely be in resting states at -100 mV with 10 μm [Ca^2+^]_in_. We repeated nP_O_ measurement at three negative potentials (-100, -120 and -140 mV) with 10 μm [Ca^2+^]_in_ and found that the change of nP_O_ induced by extracellular acidification was not significantly different at these voltages. The new result is included as a new panel in Figure 2 (Figure 2B).

Introduction, "[…] whether BK channels is sensitive" should read "whether BK channels are sensitive"

Corrected.

Also "Given the possible presence of BK channels […]" should probably read "Given the presence of BK channels", unless the authors have reason to doubt their presence at these loci. Same for "Given the possibility that BK channels may be expressed".

Even though we have no direct evidence to argue against the presence of BK channels at these loci, our result does indicate that the activity of BK channels should be highly suppressed due to the extremely low extracellular pH at these loci. Therefore, we want to be cautious with these statements. Furthermore, there may be still some doubt about whether BK channels are really present in such membranes. For example, extensive proteomics of mitochondria have failed to reveal any Slo1 protein.

Reviewer #3:[…] I do, however, have some questions and a suggestion for a point of discussion. While the apparent IC50 of inhibition at pH 3.8 does implicate acidic residues, the pH-dependence is very shallow with a hill coefficient of 0.41, such that shifts in activation are observed even at pH 7 or 6 (relative to pH 9). The mechanism underlying this shallow pH-dependence is not addressed, but is interesting because it makes the effect potentially relevant to physiological or pathophysiological pH conditions other than the extreme example of lysosomes. The shallow pH-dependence would seem to suggest either that (1) the 3 aspartic acids identified may have very different pKa's or (2) or a high pKa sensor was missed. Therefore, some relevant questions are: Do acidic residue mutations reduce the response to pH 6 as well as pH 4 (suggesting they account for the effects near physiological pH)? and, was the effect of mutating the only external histidine in hSlo (H254) tested?Even if none of these experiments were done, I think it may be worth discussing that the broad pH-dependence could arise from D133, D147, D153 having different pKa, since this fits into the existing discussion. It is well known that the pKa of acidic residues can be altered depending on aqueous exposure and interaction with other charged residues, which the authors show are different for D133, D147, D153, and may change with voltage-sensor activation. In addition, there is previous evidence to suggest that D133 and/or D153 might have an unusually high pKa (at least in the resting state) because the effect of Cu^2+^ reported by Ma et al. 2008 and referenced in the discussion, was dependent on D133 and D153, but not H254, and yet exhibited a relatively high apparent pKa of 6.0.

Yes, the unusually shallow slope of the pH-dependent gating shift curve is an interesting observation, and as reviewer #3 pointed out, deserves further investigation to deepen our understanding of BK regulation by extracellular pH. For the current study, we added a new paragraph at the end of the subsection “Relation to previous studies and structural implications” to discuss the implication of such shallow slope based on reviewer #3’s comments.